# Salmon Louse (*Lepeophtheirus salmonis* (Krøyer)) Control Methods and Efficacy in Atlantic Salmon (*Salmo salar* (Linnaeus)) Aquaculture: A Literature Review

**Kristine Cerbule *** and **Jacques Godfroid** 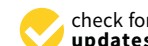

Faculty of Biosciences, Fisheries and Economics, UiT-The Arctic University of Norway, 9037 Tromsø, Norway; jacques.godfroid@uit.no

\* Correspondence: kristine.cerbule@uit.no

**Abstract:** The salmon louse (*Lepeophtheirus salmonis*) causes problems in Atlantic salmon (*Salmo salar*) aquaculture in the Northern Hemisphere, because infestations can result in both a loss of production and in fish mortality. Several types of treatment have been used to control louse infestations, but these have seen varying success. The aim of this review is to examine the efficacy and safety of commonly used treatments (chemical, biological, mechanical, and preventive measures) as documented in peer-reviewed publications. Efficacy is assessed in relation to a reduction in numbers of lice, and safety is assessed as a lack of negative treatment-associated effects on fish health and welfare (Atlantic salmon and/or cleaner fish). Most chemical treatments showed decreasing efficacy over time, together with the use of increasing concentrations as a result of the development of resistance to the treatments by lice. The need for a restrictive use of pesticides to preserve treatment efficacy has been emphasized. The use of cleaner fish was suggested to be effective, with few or no negative effects towards Atlantic salmon. The use of cleaner fish would be preferable to chemical treatment if the farmed fish health and welfare criteria are met. At present, the number of peer-reviewed publications relating to other forms of treatment and prevention are sparse.

**Keywords:** salmon lice treatment; cleaner fish; aquaculture sustainability; resistance; fish health; fish welfare; aquatic parasitic disease; peer reviewed

---

## 1. Introduction

Within their native range in the Northern Hemisphere, salmonids (genera *Salmo*, *Oncorhynchus* and *Salvelinus*) may be infested by the sea-louse species *Lepeophtheirus salmonis* and *Caligus elongatus*. This review will focus on infestations of farmed Atlantic salmon (*Salmo salar*) with the salmon louse (*L. salmonis*), as this represents a major problem for salmon aquaculture in the Northern Hemisphere. Infestation of salmon with the ectoparasitic salmon louse results in a reduced production and profitability in farming (estimated costs in Norway are around 9% of farm revenues [1]), and creates problems relating to fish health and welfare. The salmon louse has the following life stages: two non-parasitic planktonic nauplius stages; a copepodid stage; and parasitic chalimus, pre-adult, and adult stages [2]. The number of lice life stages between the infective copepodid stage to adult is reported to be two to four in the scientific literature [3,4].

Problems associated with the infestation of salmon with *L. salmonis* include reduced growth, increased mortality, increased production costs, reduced market value and consumer acceptance of the fish, and the creation of a poor image of salmon farming amongst the general public. Salmon farms may be major sources of salmon lice [5], because the holding of salmon at high densities within sea-cages

may promote the reproduction and spread of salmon lice within a farm. In areas with large numbers of salmon farms, elevated numbers of salmon lice may have a negative impact on wild migrating Atlantic salmon and sea trout (*Salmo trutta*) by reducing growth and increasing marine mortality [5].

In salmon aquaculture, a number of treatment methods have been used over the years in an attempt to combat infestations of salmon lice. Initially, chemical treatments were used, and then biological methods, involving the use of cleaner fish, were introduced to either replace or supplement the chemical treatments. Recently, other non-chemical treatments have been developed [6], including the treatment of infested fish with a warm-water or freshwater bath, mechanical delousing using brushing or water jets, and laser technology. The number of reported non-chemical delousing methods has increased since 2016 in Norway [7]; technical innovations, such as surrounding the upper layer of a cage with an impermeable skirt and the use of snorkel cages, have been introduced as preventive measures to hinder the spread of salmon lice both within and between salmon farms. Research has also been carried out on selective breeding for salmon, and on the development of functional feeds and vaccines to combat the salmon louse [8–11].

Chemical treatments used to combat the salmon louse either are added as a supplement to the salmon feed or are applied as bath treatments; these treatments comprise a range of active chemicals with different effects on the salmon louse (Table 1).

**Table 1.** Commonly used chemical compounds used to treat salmon louse infestations [6].

| Chemical Class | Active Ingredients | Effect on Salmon Louse |
| --- | --- | --- |
| **Organophosphates** | Azamethiphos | Paralysis |
| **Pyrethroids** | Cypermethrin and deltamethrin | Paralysis |
| **Avermectin** | Emamectin benzoate | Paralysis |
| **Hydrogen peroxide** | Hydrogen peroxide | Creation of gas bubbles within the body of the salmon louse, making it unable to grip a surface |
| **Benzoylurea** | Teflubenzuron and diflubenzuron | Chitin synthesis inhibitor—the louse cannot moult successfully |

The efficacy of a chemical treatment is lessened at high fish densities, with high intensities of treatments, and by the development of resistance to treatments by lice over time [12]. Increased fish density leads to a higher host availability, which can result in a higher parasite abundance [6,13–15]. Concerns have been raised about potential negative impacts of chemical treatments on farmed fish and non-target species that may occur in the area close to the aquaculture facilities [16].

Cleaner fish may feed on salmon lice and remove these ectoparasites from farmed salmon. The cleaner fish are held in sea-cages along with Atlantic salmon, and are thus given the opportunity to prey on the salmon lice that are infesting the salmon. Cleaner fish include the lumpfish (*Cyclopterus lumpus*) and several species of wrasse, such as ballan wrasse (*Labrus bergylta*), goldsinny wrasse (*Ctenolabrus rupestris*), and corkwing wrasse (*Symphodus melops*). As a result of their tolerance to low water temperatures, lumpfish are preferred as cleaner fish in colder regions. The potential transmission of diseases between the salmon and the cleaner fish, and vice versa, is one concern in using this method to combat the salmon louse [17,18]. Reliance on cleaner fish wild catch is unsustainable and needs to be addressed [19].

Other non-chemical treatments, such as warm-water or freshwater baths, exploit conditions that the sea louse is unable to tolerate, but that are tolerated by the farmed salmon. However, some negative effects have been observed on the health and welfare of farmed salmon [6]. Damage can be inflicted on the fish during transfer to and from the treatment tanks. Preventive, technological methods are being developed to reduce the risk of infestation of farmed salmon by the salmon louse. Such methods include the enclosure of the upper layer of the sea-cage within an impermeable skirt (blocks particles within the upper part of the water column, shielding salmon from lice exposure [20]), snorkel cages (aimed to keep salmon deeper in the water column and allowing access to the surface waters by

an enclosed tube [21]), and the use of submerged lighting and feeding at depth to attract salmon to deeper depths.

Infestation of salmon with salmon lice from areas close to farms tends to cause more severe symptoms (enhanced virulence) to the fish than does the infestation with lice collected from a wild population far-removed from salmon farms [22]. The problem is, therefore, primarily to develop effective treatments that do not select for resistance.

The aim of this review is to analyse research relating to the measures taken to control salmon louse infestations in Atlantic salmon aquaculture, i.e., chemical treatments, use of cleaner fish, mechanical treatment, and preventive methods. Cited works are restricted to those published in the international, peer-reviewed scientific literature; this approach was chosen to comply with measures that have been put into place to safeguard research and publishing ethics.

Efficacy of a treatment, as defined in this review, refers to reductions in the number of salmon lice, and safety is defined as an absence of any negative effects on the health and welfare of the salmon (and cleaner fish, if used) associated with a treatment.

## 2. Results

A search for publications in "Scopus" and "Web of Science" databases on 01 September 2019 yielded 366 publications, and after screening, 75 publications remained as possible sources of information for use in this review. Drawbacks identified in the excluded publications included a lack of clarity with regard to data collection and analysis, and/or incomplete information given about the methods and results of statistical analysis.

Of the 75 selected publications, 52 assessed the efficacy of chemical treatments on salmon lice, 14 discussed the efficacy of using cleaner fish intervention, and nine described mechanical or preventive measures to combat the salmon louse.

### 2.1. Chemical Treatments

The efficacy of chemical treatments has changed over time since their introduction, with efficacy generally tending to decrease with the passage of time (Table 2), probably as a result of the development of resistance in populations of the salmon louse.

**Table 2.** Efficacy of chemical treatments in relation to time since their introduction.

| Treatment Group | Date of First Publication Describing Treatment | Efficacy of Treatment Reported in the First Publication (Concentration) | Year When Resistance to Treatment First Documented | Efficacy Reported in Most Recent Publication (conc.) |
|---|---|---|---|---|
| Hydrogen peroxide | 1993 | 20% pre-adult survival (1.5 g/L bath) [23] | 1994 | 16%–50% pre-adult survival (1.5 g/L bath) [24] |
| Pyrethroids | 1998 | 50% immobilization of lice (1.03 µg/L bath) [25] | 2001 | immobilization for resistant lice strains 13.2%–20%, and 70.3%–80% for sensitive lice strains (2 µg/L bath) [26,27] |
| Benzoylurea | 1995 | 69.4%–77.5% effective (10 mg/kg fish biomass) [28] | 2015 | 96% reduction (700 µg/L bath) [29] |
| Organophosphates | 1996 | 100% gravid female reduction; 98.3% pre-adult reduction; 68% chalimus reduction (100 µg/L bath) [30] | 2012 | 19.1% immobilization of resistant strain (100 µg/L bath) [26] |
| Avermectins | 1999 | 68%–98% immobilization of lice (50 µg/kg fish biomass) [31] | 2019 | no significant difference between the control and treatment groups (50 µg/kg fish biomass); significant difference (1.2 in lice/salmon compared with 3.9) between the control and treatment group (150 µg/fish biomass) [32] |

For all chemical treatments, there have been reports of the development of resistance at some point after the introduction of the treatment (Table 3). For benzoylurea, treatment resistance was

not reported until 2015 [33], and there is clear evidence for resistance formation in the efficacy of the hydrogen peroxide treatment over time [23,24] (Table 2). There was insufficient evidence to enable an association between each particular chemical treatment and resistance to be established (Fisher's exact test, df = 4, $p$ = 0.116).

**Table 3.** Reports of development of resistance to chemical treatments, analysed by the number of publications.

| Chemical Treatment | Resistance Recorded (Number of Publications) | No Resistance (Number of Publications) | Not Mentioned | Percentage of Publications Reporting Resistance |
|---|---|---|---|---|
| Hydrogen peroxide | 5 | 3 | 1 | 55.5% |
| Avermectins | 19 | 8 | 1 | 67.8% |
| Pyrethroids | 5 | 1 | 2 | 62.5% |
| Organophosphates | 4 | 1 | 0 | 80.0% |
| Benzoylurea | 1 | 5 | 0 | 16.7% |

### 2.2. Cleaner Fish

Of the 14 publications reporting the use of cleaner fish to combat the salmon louse, five reported on the use of wrasse (goldsinny wrasse, ballan wrasse, and corkwing wrasse), whereas nine (all published since 2014) reported on the use of lumpfish. Various stocking densities of cleaner fish were used in the trials, ranging from 4% to 15% (cleaner fish to salmon numbers). In early research with wrasse, the stocking densities of the cleaner fish were high to compensate for large losses of wrasse from sea-cages (e.g., 200–300 wrasse could be lost each week per site (six cages)) [34].

The stomach content analysis carried out on wrasse species used as cleaner fish indicated that the ingestion of salmon lice varied widely, ranging from 7 to 46 lice per wrasse on average [35]. For lumpfish, the percentage of fish that had ingested salmon lice varied from 15% to 38%, but there was no information given about the numbers of lice present within each stomach [36]. Lumpfish are most effective at removing mature female salmon lice from salmon, with a 97% decrease having been reported [37]. In general, small lumpfish seem to be more effective grazers on salmon lice than larger conspecifics [38]. In all of the publications examined, the average number of salmon lice was reported to be significantly lower on the salmon held together with the cleaner fish than on those held in control sea-cages without cleaner fish [17,18,34–42].

Possible negative effects of stocking salmon and cleaner fish together in sea-cages should be examined separately for wrasse and lumpfish, because the problems associated with the use of these species as cleaner fish may differ. For wrasses, large numbers of fish disappeared from sea-cages. In one trial, in which no new cleaner fish were added over time, only 5.7% of the initial number of goldsinny wrasse and 10.2% of the initial number of corkwing wrasse were found in the sea-cages after approximately four months [39]. It was concluded that the wrasse had escaped from the sea-cages because of their relatively small body size [39]. Antagonistic behaviour was noted when Atlantic salmon were stocked together with large ballan wrasse [40]. Antagonistic behaviour has not been reported when salmon have been held together with lumpfish, but in one trial, the feed conversion ratio was lower when salmon were stocked together with large lumpfish (>350 g body weight) [41]. Lumpfish may compete with salmon for feed, and large lumpfish will probably be able to consume quite large feed pellets [17]. In one publication, some mortality of lumpfish was recorded resulting from bacterial infection with *Pasteurella* spp. [42].

### 2.3. Other Non-Chemical Treatments and Preventive Measures

Only a limited number of scientific publications covering alternative treatments and preventive measures were found in the peer-reviewed scientific literature. Most of these publications described preventive measures that could hinder the infestation of farmed salmon with salmon lice (Table 4).

There was a paucity of results relating to vaccines, submerged lights and underwater feeding, laser delousing, and mechanical removal methods reported in the peer-reviewed scientific literature.

**Table 4.** Non-chemical treatments and preventive measures used against *Lepeophtheirus salmonis*.

| Method | Number of Trials Reported in the Peer-Reviewed Literature | Significant Reduction in the Percentage of Lice Numbers Reported (Yes/No) |
|---|---|---|
| **Preventive Measures** | | |
| Selective breeding | 3 | Yes [43,44] (e.g., no treatment needed after 10 generations of selection) [8] |
| Skirt/plankton net | 2 | Yes—30%–80% reduction [20,45] |
| Functional feeds | 1 | Yes—20%–50% reduction [10] |
| Snorkel cage | 2 | Yes—72%–84% reduction [21,46] |
| **Non-Chemical Bath Treatments** | | |
| Warm water | 1 | Yes—18.6%–42% reduction [47] |
| Fresh water | 1 | Yes—81.9% reduction [47] |

Preventive measures include the use of various methods to hinder or reduce the entry of salmon lice into a sea-cage. Methods such as enclosing a sea-cage within an impermeable skirt or fine meshed netting (plankton netting), and the use of snorkel cages have been tested and have been shown to be effective at reducing numbers of lice infesting farmed salmon (Table 4). In order to assess the efficacy of such measures, environmental factors, such as currents, season, and light, must be taken into consideration [45]. There has been some research on selective breeding, functional feeds, and vaccine development, but there is little documentation relating to these and they are still at a preliminary developmental stage [15].

Other non-chemical treatment types include mechanical removal of salmon lice or the use of bath treatments, involving increased water temperatures or low salinity. Some negative effects associated with bath treatments have been noted, including stress and increased mortality of salmon, as well as environmental impacts [48]. Selection for increased tolerance of higher temperatures and lower salinities within salmon louse populations has been mentioned as a possible negative consequence of using these bath treatments [47].

## 3. Discussion

This review has focused on a synthesis and analysis of published information relating to methods used to treat and control salmon louse infestations of farmed Atlantic salmon, with considerations of their efficacy and safety.

Most research has dealt with chemical treatments. For all chemical treatments, efficacy has decreased over time, and the salmon louse has significantly increased its resistance to such treatments as a result of selection during a prolonged time of treatment [49]. Although most research has been carried out in Norway, this pattern is similar in all countries in which chemical treatments have been investigated and evaluated. Therefore, it is expected that resistance will develop if new chemical treatments are introduced, most likely via unintended selection for treatment-resistant populations of the salmon louse. For example, there are large differences in responses to treatment with the organophosphate azamethiphos treatment between sensitive and resistant strains of the salmon louse. Sensitive strains suffer almost 100% mortality, whereas in resistant strains, only 19% of the lice may be killed [26]. In addition, over time, chemical treatments have involved the use of increasing concentrations, whereas efficacy has tended to decrease, as seen particularly for pyrethroid and organophosphates. Therefore, it can be inferred that chemical treatments represent temporary measures that are reasonably effective until the development of resistance within a salmon louse population (also discussed in [15]).

All trials involving cleaner fish showed that they were effective in removing salmon lice from salmon, and that there was no evidence of a decrease in efficacy over time. On the contrary, there is a suggestion that treatments involving cleaner fish could be made more effective by selective breeding for cleaning behaviour and feeding on lice [42,50]. Cleaning behaviour performed by lumpfish or wrasse held together with Atlantic salmon can be considered as mutualism, because both species benefit from the interaction [51]. Several points relating to the use of cleaner fish to combat salmon lice should be mentioned. There were often large losses of wrasse during the early trials, with the small wrasse escaping from sea-cages. With increasing the production of cleaner fish, particularly lumpfish, in captivity, there is a concomitant reduction in reliance on the use of wild-caught cleaner fish in aquaculture. Small lumpfish are generally preferred as cleaner fish because large lumpfish may compete with salmon for feed pellets (a small reduction in salmon growth was detected) [36], and small lumpfish seem to be more effective at grazing on salmon lice than their larger conspecifics [38]. This generates a problem relating to how the lumpfish should be treated once they have reached a size at which they are both less effective as cleaner fish and may be competing with the salmon for feed. The possibility of disease outbreak in situations where salmon and cleaner fish are held together is considered a problem; cross-infections with *Pasteurella* spp. [42] and *Paramoeba perurans* [18] have been documented. The latter causes amoebic gill disease in Atlantic salmon, and this can result in significant mortalities.

There were few peer-reviewed publications describing research relating to mechanical treatment methods and preventive measures. This was unexpected, given that some of these methods are widely used within the salmon farming industry [7]. Information about these methods is available in the "grey literature", such as reports from the industry, and also in advertising material disseminated by equipment manufacturers, but the results presented in these sources have not been subjected to the rigorous assessment and quality control imposed during the peer-review process. As such, it is concluded that independent, unbiased scientific information relating to these methods for combating salmon lice is currently lacking, and that there is a research gap to be filled.

The efficacy of chemical and biological treatments used to control lice on salmon farms varies widely. At best, the treatments control the proliferation of the salmon louse and maintain levels of infestation below those that would seriously compromise salmon production and farm profitability. In addition, it is essential to look for alternatives to chemical treatments because of reduced efficacy of these treatments with time, and the possibility of negative environmental effects resulting from contamination of the marine environment with drug residues [52]. Non-chemical treatments and preventive measures, if implemented correctly, have the potential to be effective at controlling infestations, and limit the negative effects on the environment. It is concluded that the implementation of preventive measures, supplemented with the use of biological treatments when necessary, offer a reasonably effective, and sustainable, way to address the issue of controlling the salmon louse in the Atlantic salmon farming industry.

## 4. Materials and Methods

This review was prepared following the guidelines set by the Preferred Reporting Items for Systematic Review and Meta-Analysis (PRISMA; Figure 1) [53].

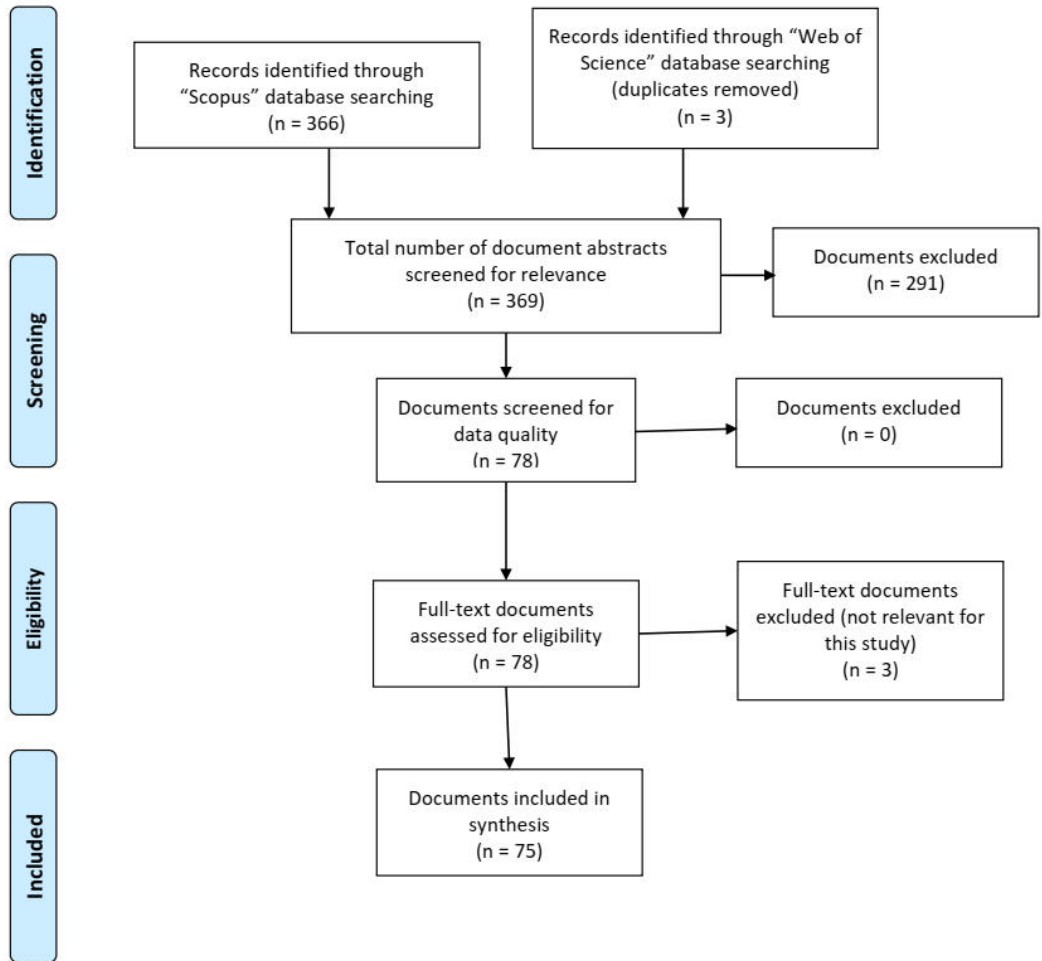

**Figure 1.** Preferred Reporting Items for Systematic Review and Meta-Analysis (PRISMA) flow diagram [53].

### 4.1. Data Collection

Multiple keywords were used as search terms in two electronic databases ("Scopus" and "Web of Science") during August 2019 and September 2019. The search included the words and phrases "salmon lice", "salmon louse", "*L. salmonis*", or "*Lepeophtheirus salmonis*" combined with descriptions of particular treatments. A complete overview of the terms used in the search is given in the Supplementary material (Table S1: Search strategy for systematic literature review, available online). The timeframe selected was from the start of 1991 until 1 September 2019, thereby covering peer-reviewed research published over a 28-year period. The search was performed by two researchers, and the studies selected for inclusion in the review were chosen on the basis of consensus.

The abstract of each publication was analysed according to the criteria given in Table 5, and all criteria had to be met for the publication to be considered for inclusion in the review.

**Table 5.** Inclusion criteria.

| | |
|---|---|
| 1. Population examined | Farmed Atlantic salmon (*S. salar*) and salmon louse (*L. salmonis*) |
| 2. Intervention method (at least one of listed) | Chemical treatment, cleaner fish, non-chemical treatment, or preventive measures |
| 3. Language | English |
| 4. Time period | 1 January 1991—1 September 2019 |

Quality control was then run on the publications to prevent the inclusion of publications that contained systematic errors or bias. The methods section of each publication was assessed according to a checklist based on Dawns and Black [54] (Table S2: Checklist for measuring document quality, available online). The checklist was adapted for this review by excluding human health-care specific questions and concentrating on examination of the quality of reporting (external and internal validity (bias and confounding) and statistical power). This checklist, with a 28-point evaluation system, has been used in previous research by adopting the following scaling: "excellent" (28–24 points), "fair" (18–14 points), and "poor" (less than 14 points) [55]. Publications rated as "poor" were excluded from this review, as they may contain flaws [55].

*4.2. Data Extraction and Analysis*

The data were extracted from the selected publications using the data extraction form (Table S3: Data extraction form, available online). The following data were extracted: methods used in the experiment/trial, characteristics of the sample, effect of treatment on numbers of lice, health impacts on the fish, and resistance of the salmon louse to the treatment. Narrative analysis was performed by tabulating and describing the data. Treatment groups were used as categories to organize the data within tables. Fisher's exact test (R-statistical package, confidence level 95%) was performed to examine for a relationship between resistance and the treatment used.

**Supplementary Materials:** The following are available online at http://www.mdpi.com/2410-3888/5/2/11/s1. Table S1: Search strategy for systematic literature review, Table S2: Checklist for measuring document quality, Table S3: Data extraction form.

**Author Contributions:** Conceptualization, J.G.; methodology, K.C. and J.G.; software, K.C.; validation, K.C. and J.G.; formal analysis, K.C.; investigation, K.C.; resources, K.C. and J.G.; data curation, K.C. and J.G.; writing (original draft preparation), K.C.; writing (review and editing), K.C. and J.G.; supervision, J.G.; project administration, K.C. and J.G.; funding acquisition, J.G. All authors have read and agreed to the published version of the manuscript.

**Funding:** This research received no external funding.

**Acknowledgments:** We would like to thank Stefano Peruzzi and Jo Espen Tau Strand for the critical reading of the manuscript. We are deeply indebted to Malcolm Jobling for the English editing and proofreading of our manuscript. The publication charges for this article have been funded by a grant from the publication fund of UiT-The Arctic University of Norway.

**Conflicts of Interest:** The authors declare no conflict of interest. The funders had no role in the design of the study; in the collection, analyses, or interpretation of data; in the writing of the manuscript; or in the decision to publish the results.

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
