# Peer review of "Salmon Louse (Lepeophtheirus salmonis (Krøyer)) Control Methods and Efficacy in Atlantic Salmon (Salmo salar (Linnaeus)) Aquaculture: A Literature Review"

_fishes, doi:10.3390/fishes5020011_

Round 1

Reviewer 1 Report

The manuscript submitted by Cerbule and Godfroid is a critical review on control methods to manage sea louse infestation of farmed salmon, focusing on their safety and effectiveness. The topic of the study is relevant, and with control approaches having changed recently, a new review on the topic area is justified. However, the analysis of the literature presented here seems at points incomplete, and contains a few errors and omissions.

In the present version, the study shows problems that need to be addressed before the manuscript can be considered for publication.

SPECIFIC COMMENTS

Abstract: “Salmon louse… is a limiting factor” - rephrase Abstract: it is stated that chemical control methods are unsustainable because of resistance formation, and that cleaner fish deployment is preferable. These statement are in the opinion of this reviewer problematic for the following reasons:
(i) Resistance formation is a classic example of evolution driven by natural selection, as discovered by Charles Darwin, which is based on descent with modification (i.e. genetic diversity of susceptibility in the population) combined with selection (overuse of the same drug, i.e. same selection pressure over many generations). Thus resistance formation is driven by overuse of one type of treatment (or of a limited number of treatment types), and not inherent to the treatment being chemical. Recommending to completely switch to one type of non-chemical treatments ignores the fact that this will introduce another selection pressure and the very real possibility that lice may genetically adapt to that pressure. At least anecdotal reports exist supporting the hypothesis that lice may possess the potential to become resistant to cleaner fish, provided continued selection pressure is applied (https://www.fishfarmingexpert.com/article/a-clear-threat-transparent-lice-cleaner-fish-cant-catch-1/)
(ii) The authors seem to have an overly narrow definition of sustainability. Cleaner fish use relies heavily on wild-caught wrasse and lumpfish. While fisheries of these species are unregulated in most countries, they are not necessarily sustainable, and may be associated to adverse ecological impacts. The same applies to drug treatments which have been linked to effects on marine non-target species. Introduction: The life cycle of L. salmonis should be briefly described, mentioning the non-feeding free-living planktonic phase preceding the host-associated parasitic phase, i.e. eggs hatch to planktonic nauplius  larvae that need to go progress to the copepodid larval stage to become infectious (which takes 2-5 days depending on temperature, during which the parasite disperses), with larvae being more prevalent in the top layers of the sea due to vertical migration. Explaining this is required to understand the efficacy of some of the control methods (e.g. snorkel cages and plankton nets). Table 1: “paralysing substance” - rephrase L54: “The chemical treatment effectiveness is negatively affected by … the increasing fish density …” It is unclear what the authors mean here. What do they mean by “increasing fish density” and why should this affect treatment efficacy? L68 “negative effects on fish health” - suggestion: “health and welfare” L74: “The infestation with salmon lice taken from aquaculture site areas tends to cause more severe symptoms to the fish than with lice taken from wild populations [14]. This suggests … lice population … becoming more resistant through …natural selection.” This statement reflects a misunderstanding of the paper cited. Virulence is not the same as drug resistance! In earlier papers (e.g. Evol Biol (2010) 37:59–67), the authors of the cited study explain that parasites infecting cultured hosts are under evolutionary pressure to shorten their life cycle, as compared to conspecifics living on wild hosts, as the farmed hosts will be harvested at an early age, usually resulting in death of the parasite (thus selection for early reproducing parasites). This trend is aggravated by control of the parasite, which would kill; them even earlier. For the parasite to mature earlier, however, it needs to extract more resources from the host (i.e. feed/grow faster), thus becoming more virulent (causing more damage). This principle applies regardless of what control method is used. Table 2: the latest publication on resistance towards a drug identified does not necessarily reports the maximum level of resistance. The authors should consider all articles available on resistance in salmon lice for the drugs considered! E.g., Carmona-Antonanzas et al 2017, (Plos One 12(7): e0180625) report that resistant strain salmon lice were unaffected by 2 µg/L deltamethrin, and that among the F1 progeny derived from crosses of female resistant and male susceptible parasites >98% of salmon lice were unaffected by 2 µg/L deltamethrin. Table 2 has further problems regarding the units (authors should use “µg/L” or “ppb” but not both) and the decimal marker (in English decimal point not comma). Table 3: The rationale behind this table is wrong. It is naïve to assume that the number of journal articles accurately reflects the prevalence of resistance. Journal articles are biased by novelty/originality: Lack of resistance (negative result) is usually not deemed worth reporting, and occurrence of resistance is worth reporting only when it is new. It is strongly recommended that the authors considered the reports from the surveillance programme for resistance to chemotherapeutants published regularly by the Norwegian Veterinary Institute which are more informative with regard to this topic. Figure 1: Timelines for the early treatments are wrong. OPs have been in use since the late 1970s, with use becoming massive during the 1980s. The authors should consider earlier reviews and articles cited therein: Roth et al, 1993, J Fish Dis 16:1-16; Burridge et al, 2010, Aquaculture 306:7-23; Torrissen et al, 2013, J Fish Dis 36:171-194. L214: “eradication of salmon lice” – No one with expertise in the area would expect it is possible to eradicate a parasite occurring on wild populations of protected fish species, unless eradication of wild salmonids is also intended! I suggest deleting the statement. L217 “both reduced effectiveness and increasing resistance” – unclear. These are not two separate issues, but two ways to express the same thing. Resistance is defined by (heritable mechanisms causing) reduced effectiveness.

Author Response

We thanks the reviewer for the critical reading of our manuscript and her/his useful suggestions.

Our response, point by point, has been uploaded in the document Response to reviewer #1.doc

Reviewer 2 Report

Manuscript Title: Salmon lice (Lepeophtheirus salmonis (Krøyer)) control method types and effectiveness in Atlantic salmon (Salmo salar (Linnaeus)) aquaculture: a systematic literature review

Manuscript Number: N/A

This paper reviews the literature on chemical, biological, and mechanical treatments to remove sea lice (Lepeophtheirus salmonis) on Atlantic salmon (Salmo salar). The authors conclude that sea lice have developed resistance to all chemical treatments and that cleaner fish may be an appropriate biological method for safely removing attached sea lice.

The conclusions based on the review (see above) are not at all surprising and they are widely known by those in the research field. As such, there is nothing novel about the review or the conclusions drawn. Still, I think the review is a valuable addition to the literature to summarize the state of the field. The authors would need to address the following concerns.

(1) Line 88. It would be nice to know up-front at this point what words were used in the literature search. These are mentioned in the Materials and Methods section (as they should be), which comes later. I suspect this is journal style and the authors have no control over it, but it is rather inconvenient to read about the Materials and Methods at the end of the paper.

(2) Table 2. In the second column heading, the last “n” in “intervention” is on a separate line.

(3) Table 2. For Pyrethroids, change “at deltamethrin” to “with deltamethrin”.’

(4) Table 2. For Organophosphates, need a space between “100” and “ppb”.

(5) Table 2. For Avermectins, need a space between “50” and “μg”.

(6) Table 2. For Avermectins, “immobilization at concentrations” of what?

(7) Line 109. It says 2016 here, but 2015 in the Table.

(8) Table 3. In the second column heading, it should be “publications”.

(9) Line 199. Wouldn’t this be “ranging from 4 to 15%”?

(10) Line 127. “The difference in lice abundance” where? Do you mean on the fish?

(11) Line 127. Change “percent” to “%” to be consistent.

(12) Line 146. “spp.” should be non-italics.

(13) Table 4. On the “Warm water” line get rid of “%” after “reduction”.

(14) Table 4. On the “Freshwater” line it should be a dash instead of a hyphen between “58” and “81.4%”.

(15) Line 161. Change “measuring” to “measure”.

(16) Line 167. Change “to” to “with”.

(17) Line 168. Add a comma after “noted”.

(18) Line 186. Change to “19.1%”.

(19) Line 191. What is meant by “decreasing tendency”?

(20) Line 193. Change “most” to “more”.

(21) Figure 2. Mention how you go from n=74 to n=51. Looks like you need a space after “=” and before “74”.

(22) Line 238. Insert “a” before “28-year period” and put in a hyphen between “28” and “year”.

(23) Table 5. By restricting the search to having at least one of the five terms in point 2, you have probably missed some key research on light traps and filter-feeding bivalves for larval lice reduction.

Author Response

We thanks the reviewer for the critical reading of our manuscript and her/his useful suggestions.

Our response, point by point, has been uploaded in the document Response to reviewer #2.doc

Round 2

Reviewer 1 Report

The review paper by Cerbule and Godfroid is much improved, but still shows a few issues.

The changes to the manuscript are overall satisfactory, except for points addressed below

Points 2-3: Comments on resistance in abstract

My comments tried to explain why I think some statements in the abstract were over-generalisations. I did not mean to suggest the authors need to introduce more topics to the abstract.

One of my critical points was that it is wrong to claim that chemical treatments are per se unsustainable because a potential to cause resistance formation. Resistance formation is a result of genetic selection, and any treatment can represent a selection pressure. Thus it is the intensity of the selection pressure, i.e. the way chemicals are used, that drives resistance formation. Other control technologies may as well select for resistance. While I agree with the authors that there is no confirmed scientific data that cleaner fish selection has caused colour changes in salmon lice, this is conceivable and anecdotal evidence exists. Ten years ago producers of certain chemicals were in denial of the possibility of resistance formation against their products using the same argument (solid data do not exist, only anecdotal reports).

I did not mean to say the authors need to consider anecdotal reports in the paper, but in my opinion they should not make sweeping statements about chemicals versus non-medicinal treatments such as those discussed above. The new version of the abstract adds another problematic general statement, namely that resistance will remain after the reduction of treatment frequency, again something that is not always true.

My suggestion is that the authors refrain from sweeping statements. This can easily be achieved by reframing existing statements to make sure reference to a specific situation made, rather claiming general validity. E.g. they could say that the fact that resistance formation against chemical salmon delousing agents was faster than the development and market introduction of new anti-sea louse drugs.

Point4 : The new statement on cleaner fish culture (Introduction L87) needs a supporting reference

Point 5 : Introduction L36: salmon louse life cycle

The introduced sentence needs language changes (“The salmon louse…”) and needs to mention the number of each of the stages, else the statement “eight” does not make sense. Consider breaking this up into two sentences.

Point 6: Table 1

Changes made are fine. One further suggestion “Chemical class” instead of “… type or substance”

Point 7: (Introduction L104-107)

I would like to repeat that virulence and resistance are distinct and unrelated problems.

 “…from salmon farms” not “… a salmon farm”

 “…which would kill them even earlier” --- unclear. Better explanation needed.

Point 12: While the adherence of the authors to a strict methodology to find literature is appreciated, the reply by the authors is overly formal and does not comment on my specific criticism, which was that the number of published papers does not provide a good representation of the resistance situation in the field, and I cannot understand why they try to use this a metric while detailed data from Norwegian field sites exist. Publication bias in Biomedicine is not the question here. I must say I am very amazed that authors affiliated with a Norwegian University suggest that a National Surveillance Program of that very country is biased and has a hidden political agenda, which raises the question of the authors’ own politics. In any case, taking advice from Matthew 7:11, the authors may want to read the reports of the said program that has collected very detailed data on the exact question they try to answer by in my opinion inadequate means, and the authors may further want to consider that some of the results of said program have been published in peer reviewed journals, e.g. Jansen et al 2016  doi:10.1371/journal. pone.0149006

Reviewer 2 Report

Manuscript Title: Salmon louse (Lepeophtheirus salmonis (Krøyer)) control methods and efficacy in Atlantic salmon (Salmo salar (Linnaeus)) aquaculture: a literature review

Manuscript Number: N/A

The authors have addressed all of my comments and suggested revisions. The paper is now acceptable for publication. The authors may wish to address the following minor revisions.

(1) Line 19. Insert “of” after “Because”.

(2) Line 21. Insert a period after “sustainable”.

(3) Line 36. Insert “The” before “salmon”.

(4) Line 37. Insert a comma after “stages” and change “places” to “place”.

(5) Line 38. Change “stage” to “stages”.

(6) Line 75. Insert a comma after “availability”.

(7) Line 89. What milestone?

(8) Line 93. Insert “the” before “health”.

(9) Line 97. Insert “column” after “water”.

(10) Line 100. Change “larger” to “deeper”.

(11) Lines 104-105. “This trend is aggravated by control of the parasites, which would kill them even earlier”. This sentence does not make sense. Please re-word it.

(12) Line 151. Change “fourteen” to “14”.

(13) Table 2. In the fifth column heading, add in “(conc.)” as in third column.

(14) Table 2. Every time there is “μ/L” it should I think be “μg/L”.

(15) Table 2. For Benzoylurea, the range should be “69.4 – 77.5%”, with no percent symbol after the first value and a dash instead of a hyphen. Also “10 mg/kg fish biomass” should be in parentheses.

(16) Table 2. For Avermectins, what is meant by groups? What groups?

(17) Line 171. Insert coma after “benzoylurea”.

(18) Lines 198-199. Poor wording of inserted section. Suggested wording: “…ranging from 4 to 15% (cleaner fish to salmon numbers)”.

(19) Line 200. Is this 200-300 wrasse lost per cage or per site? If the latter, then how many cages?

(20) Line 226. Delete “the”.

(21) Line 233. Since table captions should be stand-alone entities “Lepeophtheirus” should be spelled out.

(22) Table 4. For Selective breeding use a dash instead of a hyphen between “38” and “40”.

(23) Table 4. It should be “warm water” to agree with “fresh water”, i.e. no hyphen.

(24) Table 4. For fresh water use a dash instead of a hyphen between “58” and “81.4%”.

(25) Line 263. Insert “be” after “to”.

(26) Line 279. Insert comma after “treatments”.

(27) Lines 286-287. Data for sensitive strains should be added to Table 2, like you have for Pyrethroids.

(28) Lines 342-343. “…and the possibility of negative environmental effects resulting from contamination of the marine environment with drug residues”. This was not reviewed. The authors will need to add some references here.

(29) Figure 1. 369-290 = 79 (not 78).

(30) Line 407. Insert a space between “writing” and “original”.

(31) Line 415. Change comma to a semi-colon.

Round 3

Reviewer 1 Report

The authors have made a number of changes to the manuscript that respond to most of my queries.

I still disagree with the rationale behind table 3. The scientific literature does not follow the rationale of repeatedly looking at the same thing --- that's the job of monitoring programs. As these exist, I would find it reasonable  to consider them, but if the authors chose to do otherwise that's OK .

Regarding Table 2, I disagree with the  conclusion that there is no clear evidence for resistance formation against hydrogen peroxide. Stating that reference 24 observed only 16% of survival of parasites after treatment is a very selective reading of this study. The cited article looked at whether changing the temperature during the treatment, as compared to the ambient temperature, has a treatment benefit, and in the best case (switching from an ambient temperature of 16C to a bath temperature of 10C) this resulted in the above clearance. However, switching the temperature is not the standard procedure. The cited study used a number of other temperature courses (not changing the temperature, or different types of changes) and with conditions more typical for industry conditions (e.g. ambient temperature of 13C and no change temperature or ambient temperature of 16C and lowing this to 13C during the treatment) the survival was ~45% thus providing clear evidence for resistance formation, as has been previously described (Helgesen et al, 2015, Aquaculture Reports 1:37-42).

Author Response

Please see document attached hereafter.
